# Ano-swinMAE: Unsupervised Anomaly Detection in Brain MRI using Swin Transformer-based Masked Auto Encoder

**Kumari Rashmi**[*1]                                              EE20S051@SMAIL.IITM.AC.IN

**Ayantika Das**[*1]                                               EE19D422@SMAIL.IITM.AC.IN

**Matcha Naga Gayathri**[1]                                        EE21S048@SMAIL.IITM.AC.IN

**Keerthi Ram**[2]                                                 KEERTHI@HTIC.IITM.AC.IN

**Mohanasankar Sivaprakasam** [1,2]                                MOHAN@EE.IITM.AC.IN

[1] *Indian Institute of Technology, Madras*

[2] *HealthCare Technology Innovation Center*

**Editors:** Accepted for publication at MIDL 2024

## Abstract

The advanced deep learning-based Autoencoding techniques have enabled the introduction of efficient Unsupervised Anomaly Detection (UAD) approaches. Several autoencoder-based approaches have been used to solve UAD tasks. However, most of these approaches do not have any constraints to ensure the removal of pathological features while restoring the healthy regions in the pseudo-healthy image reconstruction. To minimize the occurrence of pathological features, we propose to utilize an Autoencoder which deploys a masking strategy to reconstruct images. Additionally, the masked regions need to be meaningfully inpainted to enforce global and local consistency in the generated images which makes transformer-based masked autoencoder a potential approach. Although the transformer models can incorporate global contextual information, they are often computationally expensive and dependent on a large amount of data for training. Hence we propose to employ a Swin transformer-based Masked Autoencoder (MAE) for anomaly detection (Ano-swinMAE) in brain MRI. Our proposed method Ano-swinMAE is trained on a healthy cohort by masking a certain percentage of information from the input images. While inferring, a pathological image is given to the model, and different segments of the brain MRI slice are sequentially masked, and their corresponding generation is accumulated to create a map indicating potential locations of pathologies. We have quantitatively and qualitatively validated the performance increment of our method on the following publicly available datasets: BraTS (Glioma), MSLUB (Multiple Sclerosis), and White Matter Hyperintensities (WMH). We have also empirically evaluated the generalization capability of the method in a cross-modality data setup.

**Keywords:** Unsupervised Anomaly Detection (UAD), Ano-swinMAE, MRI, Masked Autoencoder (MAE), Pathology Detection.

## 1. Introduction

The identification and delineation of pathology in brain MRI plays a crucial role in disease diagnosis and prognosis. Unsupervised Anomaly detection (UAD) methods alleviate the task of annotating pathologies at the pixel level. The state-of-the-art deep learning-based

---

* Contributed equally

UAD methods are Autoencoding models that learn to encode healthy data distribution. The appearance of pathological features in brain MRI scans is typically localized in certain anatomical regions. The Autoencoding networks must ensure changes are introduced in those localized pathological regions while restoring the healthy features. The changes in pathological regions consequently arise due to the fact that Autoencoders are trained on healthy data, and there is a performance degradation in these pathological regions. However, there is no constraint in the Autoencoding models that ensure pathologies will be absent in the pseudo-healthy reconstructed images. To mitigate this, a masking-based Autoencoding strategy is an intuitive approach that can be incorporated. While masking strategies have been adopted for unsupervised anomaly detection (Nguyen et al., 2021) (Iqbal et al., 2023), most of the approaches do not consider position-aware global context dependencies. These properties are capacitated by Vision transformers (ViT) (Dosovitskiy et al., 2020), which will ensure that there is an association among distant imaging features of the brain MRI scans (Wang et al., 2023) essential for in-painting the masked regions with meaningful healthy features while removing the pathological ones. However, ViT-based models lack (Xu et al., 2021)(Bietti and Mairal, 2019)inductive biases in modeling local visual structure which cannot be traded with global context since inductive biases for locality information is crucial as we can see in the case of CNN. Additionally, this limitation results in a heavy reliance on large datasets and pre-trained models, which is difficult in scarce data setups like in medical images. To mitigate these challenges, the Swin Transformer (Liu et al., 2021) introduces a shifted windows-based Multi-head self-attention (MSA) for modeling global feature relations. This not only enhances performance but also reduces model complexity. Hence, (i) we propose to utilize a Swin Transformer-based Masked Auto Encoder (Ano-swinMAE) for detecting and localizing pathologies in an unsupervised manner. We have shown the efficacy of our method using publicly available pathology datasets. Additionally, (ii) we have shown how masking helps our models map the pathological distribution closer to a healthy distribution in the latent representational space. (iii) We also validate that our method is generalizable in different datasets and computational efficiency is relatively better.

## 2. Related work

Several deep-learning approaches have been investigated in recent research on Unsupervised Anomaly Detection (UAD). Among these, methods incorporating Autoencoders (AE) and Variational Autoencoders (VAE) have proven to be effective during both training and inference (Zhou et al., 2021), (Baur et al., 2021a). Nevertheless, a common limitation observed in these approaches is the reconstructed image quality, which tends to be blurry. This blurriness poses a challenge, rendering these methods less effective for UAD tasks, as described in (Baur et al., 2021b). To overcome this limitation, researchers have worked towards utilizing the image context by adding spatial latent dimension (Baur et al., 2019), erasing spatial context, (Zimmerer et al., 2018), making use of 3D information (Bengs et al., 2021) (Behrendt et al., 2022)

As an alternative to AE, Generative Adversarial networks have been applied to the problems of UAD task (Schlegl et al., 2019). However, the unstable training nature of GANs poses challenges, leading to issues such as mode collapse and a lack of anatomical coherence (Baur et al., 2021b). Meanwhile, the recent work in UAD has utilized DDPM (Denoising Diffusion Probabilistic Model) because it exhibits scalable and stable training

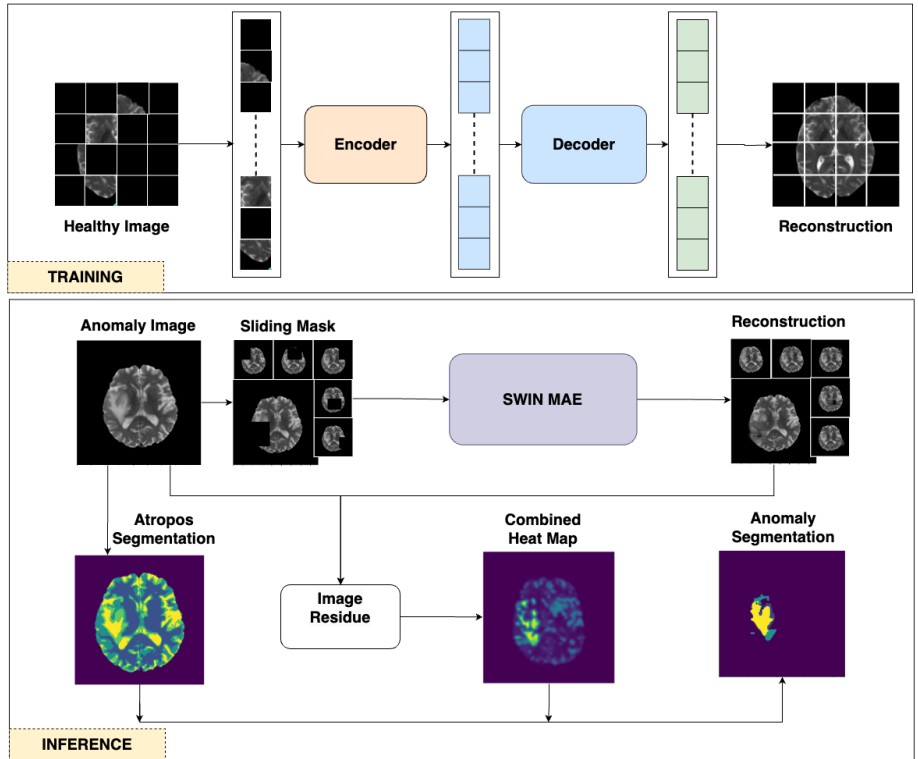

Figure 1: Main architecture consists of Training and Inference of Ano-swinMAE. During Inference, Mask slides across various parts of Anomaly brain MRI image(slice), and a pseudo healthy MRI slice is reconstructed. Then with the help of original Anomaly MRI image(slice), we get combined heatmap and one segmentation image using Atropos. With the help of combined heatmap and Atropos segmentation image, Anomaly is Localised.

properties while producing high-quality, sharp images (Wolleb et al., 2022) (Wyatt et al., 2022) (Sanchez et al., 2022) (Pinaya et al., 2022a). In the DDPM-based approach, there is a tradeoff between preserving crucial information about healthy tissues and efficiently eliminating anomalies due to the inherent noise addition strategy. Recent DDPM works that are used for UAD tasks and deal with this tradeoff are (Behrendt et al., 2023) and (Bercea et al., 2023). This tradeoff often limits the applicability of the models among diverse pathological data. Additionally, (Iqbal et al., 2023) uses a DDPM-based model which incorporates a mechanism to simulate pathology during the training phase. This strategy involves an approximation of the pathological distribution in the training phase that may not ensure capturing the true distribution.

However, the intricate structure of the brain can be captured by learning to model the relationship between individual brain structures, which can be modeled using transformer models. UAD using transformers was done by (Pinaya et al., 2022b), (Ghorbel et al., 2022). The transformer-based approaches incorporate masking(Zhou et al., 2023) strategies that ensure the mapping of the pathological brain into a healthy distribution without adding any additional constraints. Employing transformer-based Masked Autoencoders (MAE) (Hu et al., 2023), (Yu et al., 2022), (Georgescu, 2023) generate a pseudo-normal counterpart of

the normal image. Additionally, (Georgescu, 2023) integrates a pseudo-abnormal module to simulate pathology to train a classifier to discriminate between healthy and unhealthy. Their dependency on the secondary classifier clearly highlights that MAE independently is not suitable for pathology detection.

## 3. Methodology

### 3.1. Problem Formulation

Let $x \in \mathbb{R}^{H \times W \times C}$ be a healthy or pathology brain MRI scan with dimensions $H \times W$ and C channels. The task of unsupervised detection and localization of pathologies in $x$ can formulated as the reconstruction of the pseudo healthy ($\hat{x} \in \mathbb{R}^{H \times W \times C}$) counterpart of $x$ and extracting the residual information by subtracting $\hat{x}$ from $x$. The reconstruction task is achieved by an Autoencoding network, which is trained to map $x$ to $\hat{x}$, where $x$ belongs to a healthy cohort input and $\hat{x}$ is its reconstruction respectively. At the inference time, $x$ belonging to a pathological distribution is given as input to the model and expected to generate $\hat{x}$ belonging to the healthy distribution.

### 3.2. Proposed Framework (Ano-swinMAE)

The overview of our proposed Swin Transformer-based MAE (Ano-swinMAE) for anomaly detection is given in Figure 1. In the training phase, an input to the model ($x$) is randomly masked at regions within the brain area, which are passed through the encoder to extract latent representations ($z$ ). These representations are further processed through the decoder to generate the reconstructed image, learning to incorporate meaningful information within the masked regions. The masking positions used during training are random but the masking ratio and size of the masks are given by us. The learning objective is a mean square error between the masked region of the original and reconstructed image.

The significance of the model lies in the integration of three key components within the autoencoding model. *First*, a Swin Transformer block (Liu et al., 2021) is introduced within the model along with a patch merging layer which reduces the number of positional information (token). This is quite essential for task which faces data scarcity and requires lesser model complexity. *Second*, A windowing strategy is incorporated which limits the capability of every patch to be masked. This strategy allows patches within a certain window size to be masked. This masking strategy resembles the pathological occurrence since frequent masking with a small mask size will fail to reconstruct in realistic larger pathological cases. *Third*, The token of a masked patch is allowed to pass through the model such that the model is aware of the absolute positions of all the patches. This strategy helps in obtaining reconstructed images with better fidelity. We have incorporated all these three strategies and adopted the architectural details of (Dai et al., 2023) to build the encoder-decoder structure of the Ano-swinMAE.

In the inference phase, we generate a non-overlapping sliding mask sequentially moving across regions having brain pixels as indicated in the Inference block of Figure 1. Each masked reconstruction undergoes L1 norm-based computation to derive a residual map compared to its original counterpart. These individual residual maps are then aggregated to form a coarse anomaly map.

## 4. Experiments and Results

### 4.1. Dataset Description, Pre and Post Processing, Evaluation Metrics and Implementation Details

**Dataset Description**: We have used four publicly available datasets, IXI (IXI), BraTS21 (Baid et al., 2021), MSLUB (Lesjak et al., 2018) and WMH dataset(Kuijf et al., 2022). The IXI dataset(T1, T2) consisting of 580 subjects MRI volume is used as a reference for training our healthy distribution. BraTS21 dataset(T2) consisting of 1251 subjects with MRI volume, and the MSLUB dataset (T2) consisting of 30 subjects with MRI volume, are used for evaluating the performance of our model with different baselines. WMH dataset(Flair) consists of 100 subjects with MRI volume, and the BraTS21 dataset (T1, T2) are used for evaluating the generalizability of the model.

**Pre and Post Processing**: In order to standardize the images over geometric variations, we perform skull-stripping using HD-Bet (Isensee et al., 2019) and rigid body registration with SRI24 (Rohlfing et al., 2010) atlas, effectively resulting the volume to be of 240 $\times$ 240 $\times$ 155 dimensions. Additionally, to mitigate photometric variations, we perform bias field correction and normalization to the [0,1] intensity range. For post-processing, morphological filters are applied to the coarse anomaly map given by the aggregated residual output of our model to eliminate smaller objects, followed by a connected component-based analysis. This analysis isolates the significant residual components. A Gaussian mixture model-based approach is employed on the original image through Atropos [1] to generate a segmentation mask. Integrating this Atropos segmentation information refines the anomaly map, creating precise segmentation boundaries of the pathologies.

**Evaluation Metrics**: For quantifying the segmentation performance of different models, we have considered the standard definitions of Dice coefficient and Area Under the Precision Recall curve (AUPRC). For analyzing the latent representations of our model, we have projected the high dimensional vectors in a 2d space by using Umap projections (McInnes et al., 2018). In order to further analyze the spatial relationship of the data, we have performed k-means clustering of the 2d points and extracted the eigenvectors and eigenvalues of the covariance matrix of each cluster. These eigen-components are used to draw an ellipse to represent the spread and orientation of each cluster.

**Implementation Details and Baselines:** Models are implemented in PyTorch 2.0.1 version on an 80GB NVIDIA A100 GPU and CUDA Version: 12.1. For every step mentioned in the training algorithm, models are trained for 400 epochs using the Adam optimizer, with a learning rate of $1e^{-3}$. We have compared our model performance with several existing baselines, such as VAE (Baur et al., 2021b), f-AnoGAN (Schlegl et al., 2019), MAE (He et al., 2022) and autoDDPM (Bercea et al., 2023).We have evaluated on all the baseline methods by adopting the existing implementations. The details of the parameters used in post-processing are mentioned in Appendix A.1. The code for our proposed method will be available at Ano-swinMAE repository [2].

---

1. https://antspyx.readthedocs.io/en/latest/segmentation.html

2. https://github.com/rashmi05pathak/Ano-swinMAE

## 4.2. Results and discussion

### 4.2.1. Quantitative And Qualitative Analysis

The quantitative evaluations of our method against baselines are summarized in Table 1. Our model, Ano-swinMAE, shows an increment in performance compared to the other baselines. The effectiveness of different mask sizes in our model varies depending on the dataset: a mask size of 32X32 performs better for MSLUB, whereas a mask size of 64X64 yields superior results in BraTS21. The performance drop of autoDDPM compared to our method could be due to the noising strategy employed by autoDDPM to map pathological distributions to healthy ones. This strategy might not consistently prevent the reconstruction of pathological regions, such as hyper-intense large high-grade gliomas found in the BraTS21 dataset. MAE could not achieve performance increments because of its reliance on a large amount of data, leading to lesser image fidelity compared to Ano-swinMAE. GAN-based approaches often lack one-to-one mappings in healthy brain regions due to modeling challenges like mode collapse. Similarly, VAEs suffer from posterior collapse, resulting in low-fidelity images and an increased occurrence of false positives. Furthermore, Table 1 highlights that our model exhibits better inference time when compared with baselines that have relatively good Dice and AUPRC.

Table 1: Comparison of Ano-swinMAE with other baseline models which are used for unsupervised pathology detection in brain MRI.Ano-swinMAE(32x32) uses a 32x32 mask size in brain MRI and shifts by 32 pixels.Ano-swinMAE(64x64) uses a 64x64 mask size in brain MRI and shifts by 64 pixels.

| Model | BraTS21 | | MSLUB | | Parameters(M) | Inference Time(s) (per MRI slice) |
|---|---|---|---|---|---|---|
| | Dice[%] | AUPRC[%] | Dice[%] | AUPRC[%] | | |
| VAE(Baur et al., 2021b) | 31.11 | 28.80 | 6.89 | 5.00 | 4.96 | 0.02 |
| f-AnoGAN(Schlegl et al., 2019) | 24.16 | 22.05 | 4.18 | 4.01 | 5.56 | 0.04 |
| MAE(He et al., 2022) | 31.97 | 25.44 | 15.46 | 10.56 | 329 | 90 |
| autoDDPM(Bercea et al., 2023) | 35.80 | 29.07 | 19.35 | 11.79 | 18.50 | 16.32 |
| Ano-swinMAE(32x32) | 42.55 | 30.44 | 19.78 | 12.78 | 26.10 | 52 |
| Ano-swinMAE(64x64) | 42.92 | 30.24 | 18.52 | 12.50 | 26.10 | 14.38 |

The pathology segmentation masks produced by autoDDPM, MAE, and our method for BraTS21 T2 data are illustrated in Figure 2. We present results across three distinct intensities in pathological regions relative to those in the ventricles and sulci (hyper, medium, and low). Our method and MAE exhibit a consistent trend in performance across these intensity levels, with metrics decreasing in the order of hyper, medium, and low-intensity cases. In contrast, autoDDPM shows a reverse performance trend. Our model demonstrates improved anomaly capture compared to MAE, due to the blurrier reconstruction of MAE that generates more false positives. The autoDDPM model tends to reconstruct traces of pathologies in hyper-intense regions, leading to more false negatives. Additionally, the integration of Atropos-based segmentation information enhances the precision of anomaly boundaries.

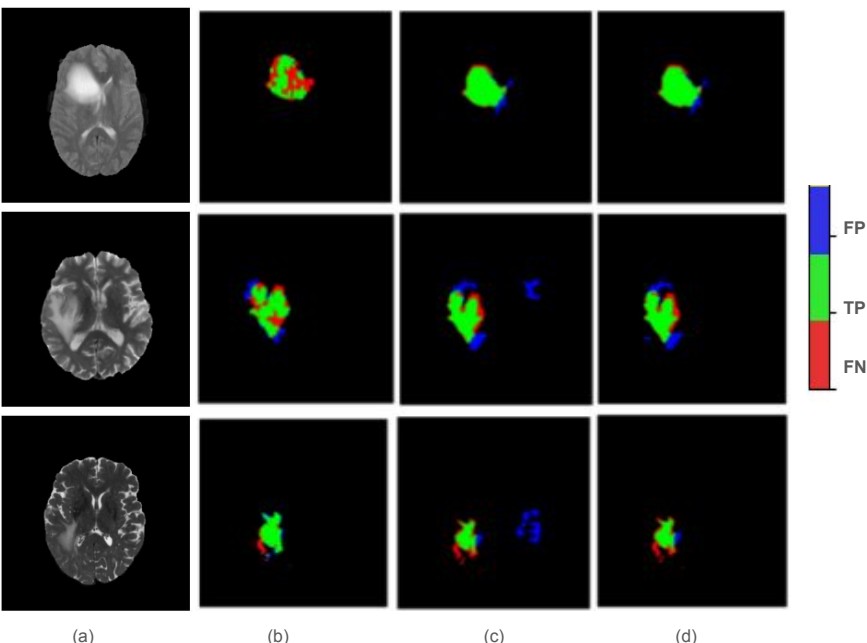

Figure 2: (a) represents original BraTS21 (T2) brain MRI, (b) represents the pathology segmentation from the autoDDPM method, (c) represents pathology segmentation from the MAE method, and (d) represents the pathology segmentation from our Ano-swinMAE method

Table 2: Generalization study of Ano-swinMAE (64X64) and AutDDPM models. We have utilized WMH flair brain MRI slices for this study along with BraTS21. Our model Ano-swinMAE(64X64) and the autoDDPM models trained with IXI dataset are used here

| Model | Training on IXI Data | Inference Data | | Dice[%] | AUPRC[%] |
|---|---|---|---|---|---|
| AutoDDPM | T1 | BraTS21 | T2 | 30.87 | 25.67 |
| | T2 | BraTS21 | T1 | 29.77 | 25.88 |
| | T1 | WMH | Flair | 35.97 | 32.91 |
| Ano-swinMAE | T1 | BraTS21 | T2 | 38.99 | 27.19 |
| | T2 | BraTS21 | T1 | 28.62 | 25.11 |
| | T1 | WMH | Flair | 35.92 | 29.95 |

#### 4.2.2. ANALYSING THE EFFECT OF MASKING IN THE LATENT SPACE

In Figure 3, we compare the latent space of masked and unmasked images. From Figure 3 (a), it is evident that the 2d UMAP projections of the latent representations of Ano-swinMAE tend to separate into two clusters when healthy data slices from IXI and pathological ones from BraTS21 are sent to the model without masking. Each of the ellipses formed from the covariance matrix of each cluster approximately contains projected points from one of the datasets (IXI or BraTS21). Figure 3 (b) indicates that the projections of both datasets tend to collapse within a single cluster. This indicates that since the encoding capability of the model did not explicitly enforce any constraint to map pathological data

into a learnt healthy distribution but the masking strategy has enabled the mapping. Consequently, this allows the latent representational space to encode diverse semantics, unlike modeling capabilities like GAN and VAE, which constrain the latent space to follow a prior standard Gaussian distribution.

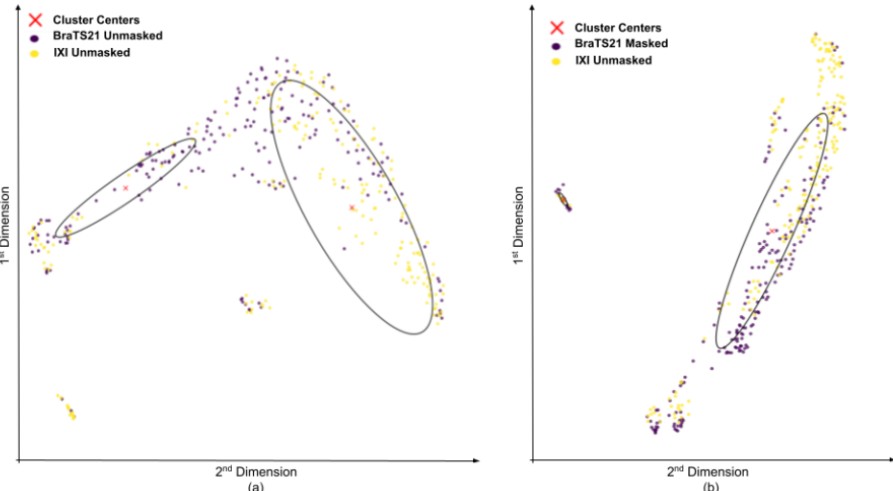

Figure 3: Two-dimensional UMAP-based projection of the latent space vectors of the Ano-swinMAE model. From left to right: (a) displays the UMAP projection of latent vectors of both IXI and BraTS slices without masking, and (b) displays the projection of BraTS slices, with masking applied to pathological locations alongside unmasked IXI slices.

### 4.2.3. GENERALIZATION ACROSS MODALITIES

From Table 2, it is evident that our model Ano-swinMAE shows incremental performance when evaluated on a cross-data setup. When the model is trained on IXI T1 data and evaluated on BraTS21 T2 data, it can perform better than the baseline under a similar setup. Similarly, when IXI T2 is used for training and BraTS21 T2 data for inference, the quantitative metrics are better for our method. In the case of WMH Flair data, our model has slightly inferior results to the baseline. Figure 3 (a), also supports that our model has better Generalization capability since the representational space encodes meaningful semantics tending to form separable clusters for anomaly and healthy data. Ano-swinMAE exhibits better performance when trained on T1 and evaluated on T2 since T2 enhances the pathological appearances in the images and it is easily discriminative. Whereas in the case of autoDDPM if the pathological information is evident, then it is present after the noising process. Further details are discussed in A.3.

## 5. Conclusion

In this work, we have proposed Ano-swinMAE, a transformer-based system for unsupervised anomaly detection, which further extends the scope of transformer usage in the field of medical imaging. Our method has outperformed the baselines as well as has shown promising results for generalizable capability. Through latent space analysis, we have observed that masking is quite effective for pseudo-healthy reconstruction of brain MRI.

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

## Appendix A.

### A.1.  Inference and Post-Processing Details

During inference on pathological data, the non-overlapping sliding mask ($64 \times 64$) moves across 6 unique positions to cover all brain regions within the image. We have used L1 normalization on the residual images, obtained by subtracting the reconstructed images from the unmasked original image for each of the 6 positions. All the reconstructed images are normalized (0 to 1 normalization) and added to form a combined coarse anomaly map. The results of the post-processing steps are given in Figure 4. The combined map is eroded with kernel size ($k \times k$) to remove very small false positives and then it is thresholded to retain pixel values greater than 0.5. For BraTS21 $k = 3$ and for MSLUB & WMH $k = 1$. To extract the unique objects in the binary map, we perform connected component analysis. Each of the unique objects is assigned a mean intensity value acquired from the combined map before thresholding. Depending on the higher mean intensity of the entire object area, a certain number of objects are retained. The number of objects that have given the best result for BraTS21 is 2, and for MSLUB & WMH, it is 7 since BraTS21 primarily has a few larger appearing pathologies and the other two sets have multiple smaller pathologies. The obtained map is dilated with kernel size ($3 \times 3$) to obtain a filtered combined map.

We have performed Atropos-based multi-class segmentation (the number of classes is 5) on the original image. We also obtain the unique objects from these segmentation masks and look for objects that share maximal area overlap with the filtered combined map. This gives us the final segmentation map, maintaining the same number of objects as the filtered combined map while refining the precise object boundaries obtained from the Atropos segmentation.

### A.2. Visualisation of Attention maps

To compute the attention map across the encoder layers of Ano-swinMAE, we extracted the feature vector outputs from each encoder layer and then computed the mean across channels for each feature vector output. For instance, the feature vector output dimension from encoder layer 1 is 56x56x96; after averaging across 96 channels, a 56x56 attention map is obtained.

In Figure 5, Rows (ii) and (iv) reveal that the attention (yellow region i.e. more intensity region), is concentrated within the masked and pathological regions of the input image, as indicated by (ii)a and (iv)a. As we progress through the encoder layers, attention becomes more dissipated. A similar pattern is seen when analyzing the attention map in a 64x64 masking setup, as depicted in Figure 6. This model behavior can be attributed to the training data consisting of healthy MRI slices; when presented with unhealthy slices, the model

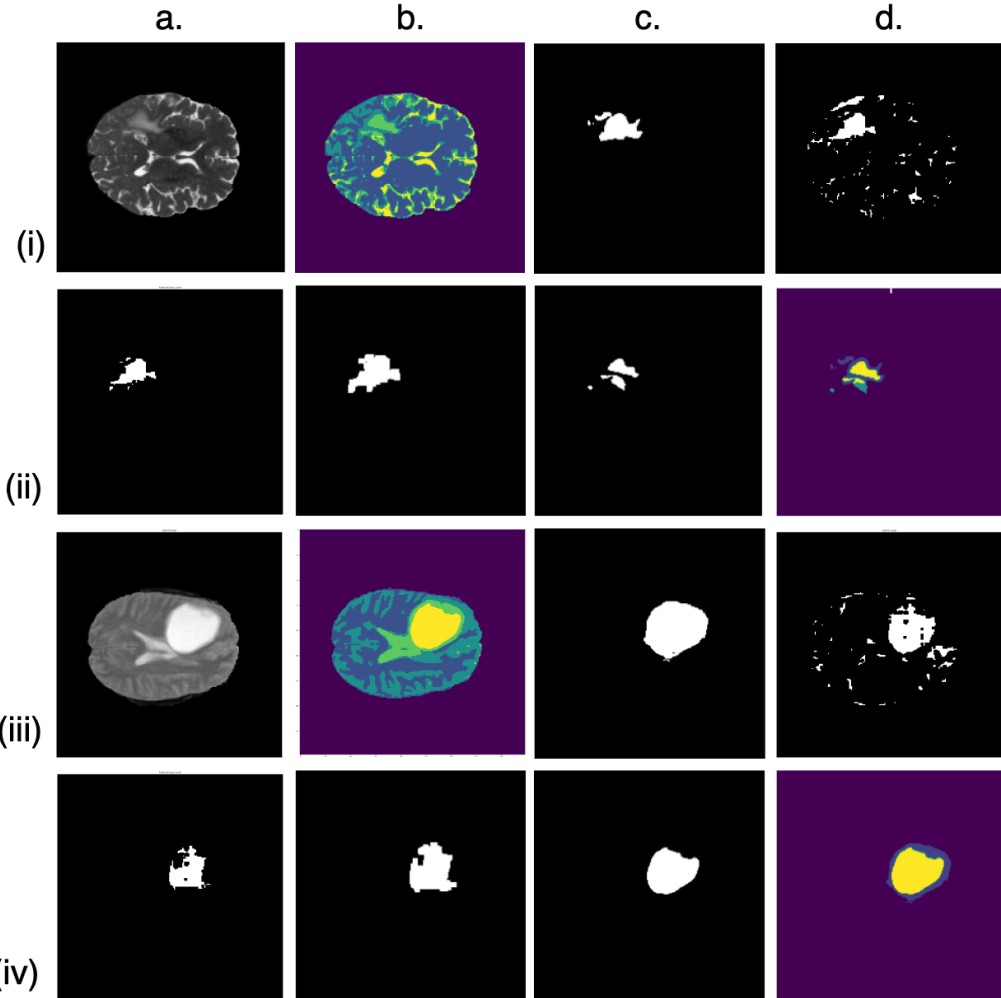

Figure 4: Post-processing steps Rows (i) and (iii): column-wise a. Original Image b. Atropos Segmentation c. Ground Truth d. Combined map after Morphological operations. Rows (ii) and (iv): column-wise a. Combined map after Connected Component Analysis b. Filtered Combined map c. Filtered Combined map with Atropos d. Overlay of Ground Truth and Prediction.

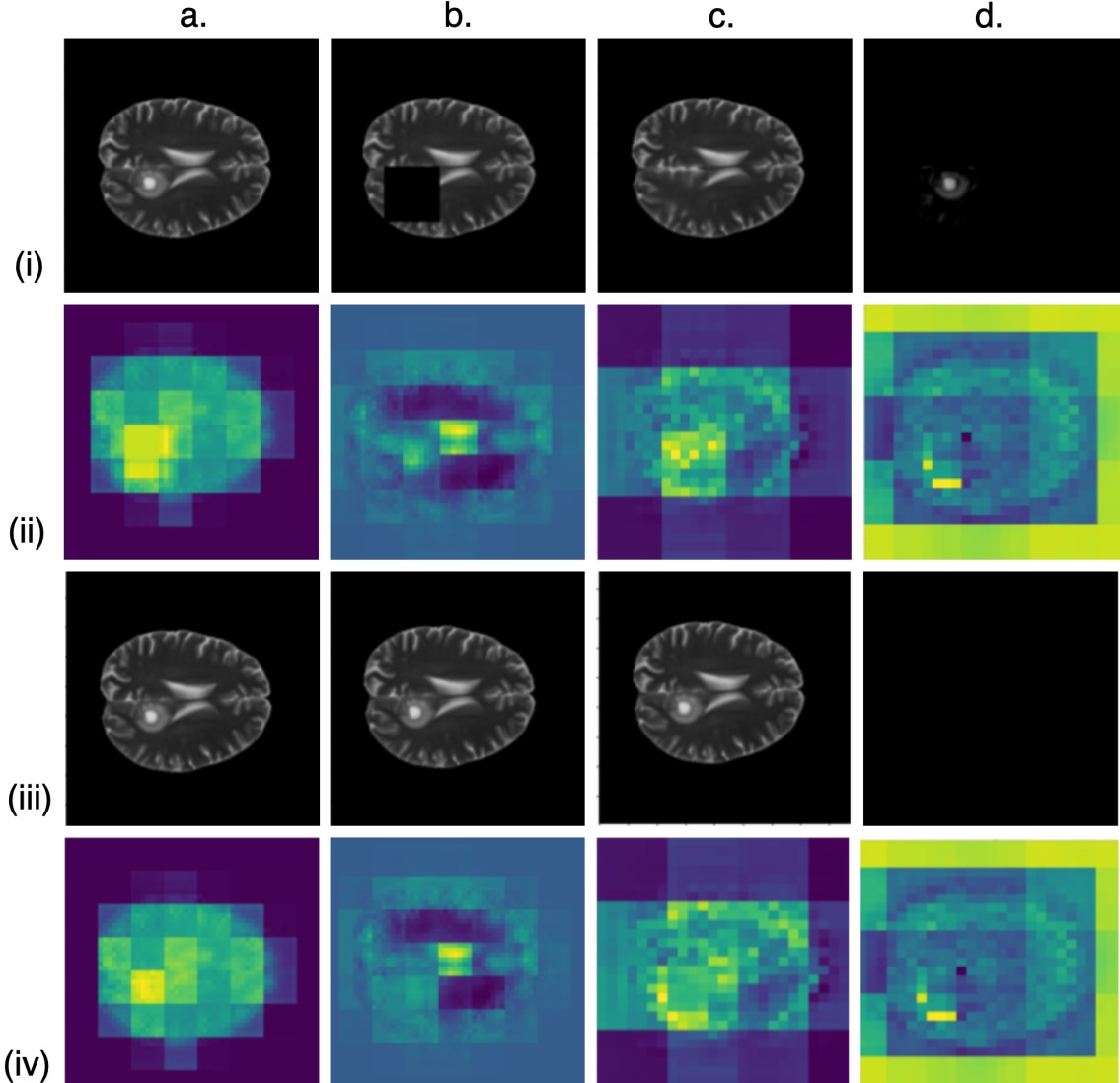

Figure 5: Visual Results of attention map of 32x32 mask setup. Row(i) and Row(iii) column-wise a. Original Image b. Masked/Unmasked Input Image c. Reconstructed Image d. Residual Image. Row(ii) and (iii) are the layer-wise Attention Maps for Masked and Unmasked Input Images respectively.

attempts to reconstruct their healthy counterparts. Furthermore, the model demonstrates increased attention when the pathological region is masked.

## A.3. Generalization Across Modalities

T2-weighted MRI highlights pathologies like tumors with high water content or those surrounding edema. Hence pathologies mostly become more visually discriminable in T2 modality. Visual results of cross-modality setup are shown in Figure 7. When, we are assessing the metrics for T2 while being trained on T1, we observe that the performance

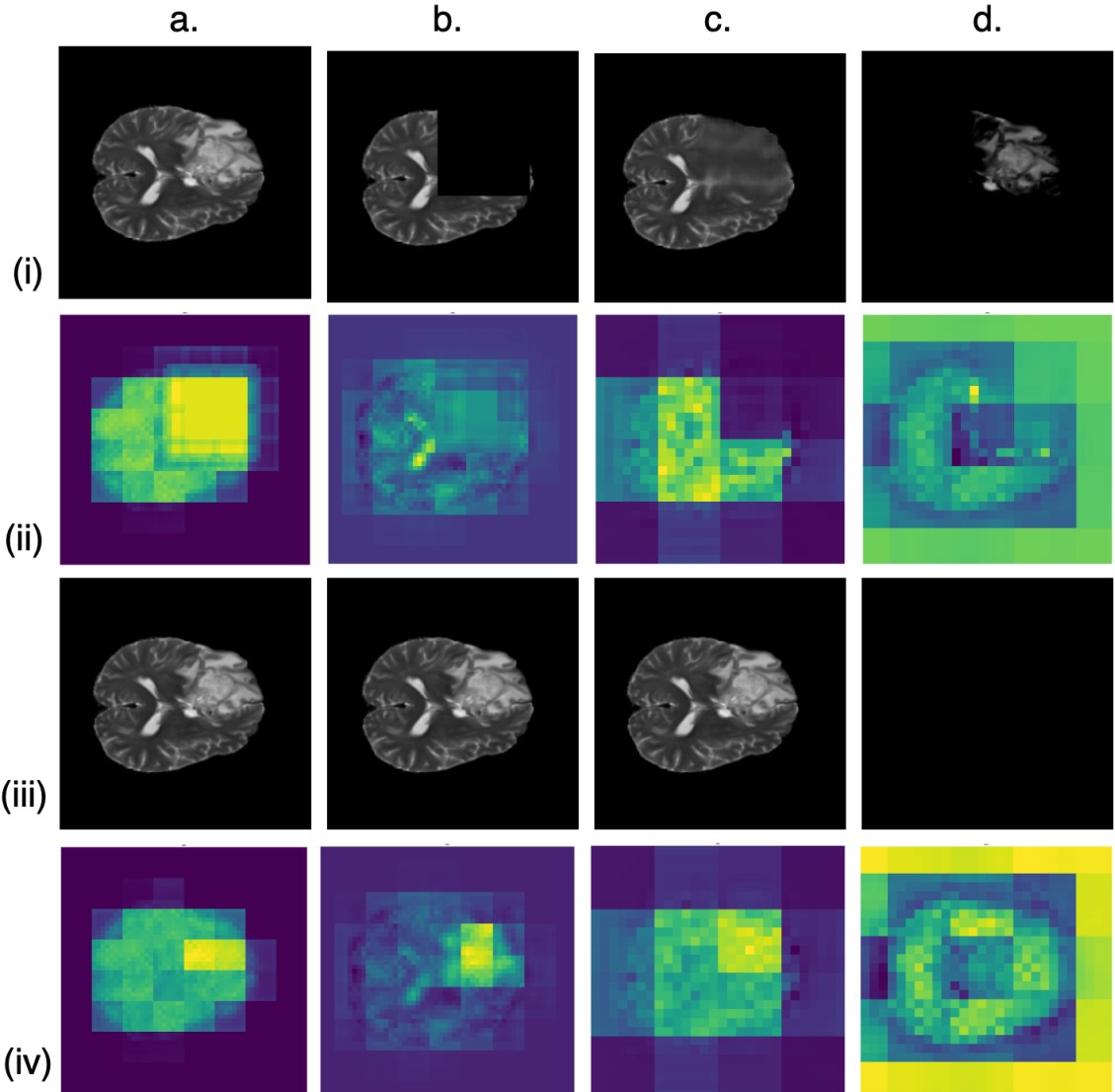

Figure 6: Visual Results of attention map of 64x64 mask setup. Row(i) and (iii) column-wise a. Original Image b. Masked/Unmasked Input Image c. Reconstructed Image d. Residual Image. Row(ii) and (iii) are the layer-wise Attention Maps for Masked and Unmasked Input Images respectively

of Ano-swinMAE is better since the model is able to easily differentiate between healthy and pathological content. When the modalities are reversed (inferring on T1 while trained on T2) due to the lesser discriminative appearance of the pathology, the performance of the model drops. There is an analogy between the appearance of the pathology and the performance of the model since it attempts to fill the masked regions with contextual neighborhood information. Whereas in the case of autoDDPM the trend is reversed since having a more evident appearance of pathology (as in T2), the chances to pass the pathological information through the noising process is higher. This degrades the performance scores.

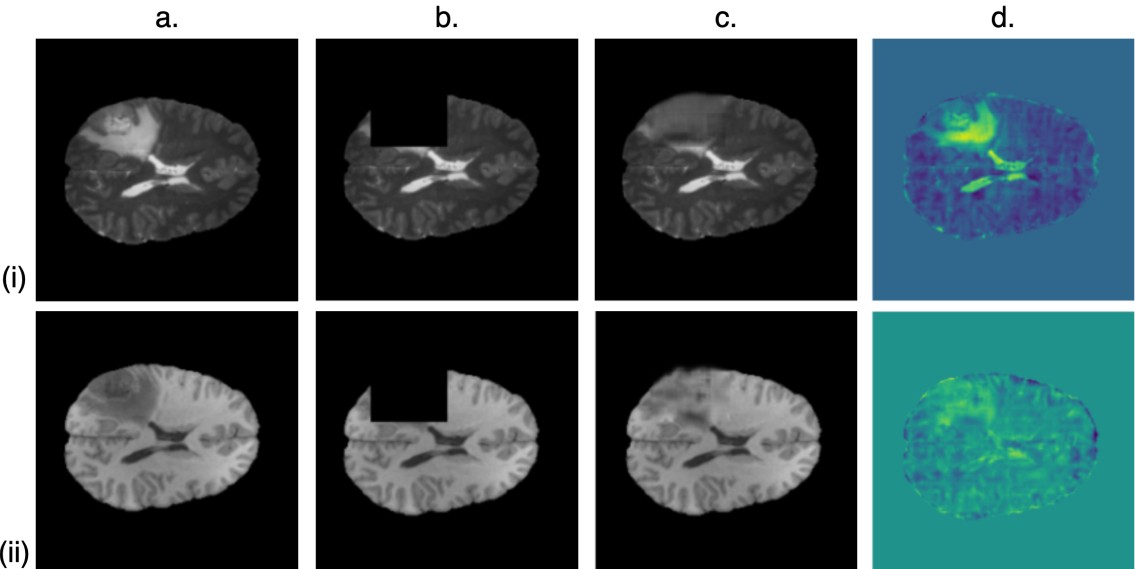

Figure 7: Visual Results of cross-modality setup. Row-wise (i) and (ii) have the result of T2 and T1 images, while the models were trained on T1 and T2 images respectively. Column-wise a. Original Images b. Masked Images c. Reconstructed Images d. Residual Images

