# OpenReview forum: "Ano-swinMAE: Unsupervised Anomaly Detection in Brain MRI using swin Transformer based Masked Auto Encoder"
_MIDL.io/2024/Conference — MIDL 2024 Poster_

### Official Review · Reviewer_sB1L · 2024-02-29

**Confidence:** 4
**Preliminary Rating:** 4
**Final Rating:** 4

**Summary:**

The authors address the difficult task of unsupervised anomaly detection in human brain MRI. To tackle this issue they propose a masking auto-encoder approach based on a Swin transformer architecture to contraint the model to remove pathological features when reconstructing a pseudo-healthy version of the patients' MRI, while not deteriorating the healthy regions. The model was trained on a dataset with healthy MRI only, and then the inference and evaluation was performed on three publicly available MRI datasets from patient with Glioma (BraTS), Multiple Sclerosis (MSLUB) and White Matter Hyperintensities (WMH).

**Strengths:**

The proposed manuscript has several strengths, among which the use of the multi-head self-attention from Swin Transformer to model the global feature relations : the authors do the hypothesis that masking, and its inpainting counterpart task, may be boosted by the model if it learns such global feature relations during training, i.e., the target at leveraging the inpainting position dependency, which seems a promising proposition. To study the effect of the proposed masking strategy, the authors projects the latent space representations of samples from the different datasets into a reduced dimensional space (using 2d UMAP) : they shows a strong alignment of the IXI Unmasked and the masked BraTS21 distributions, supporting that the model removed (or at least did not encode) typical patterns of pathological regions in the latent space when using the proposed masking approach. Finally, the presented results are obtained on three publicly available dataset which increase the contribution reproducibility. The associated code is not yet available, but it is announced to be promptly released (upon acceptance ?).

**Weaknesses:**

The meta parameter defining the mask shape is critical as it impacts the results and inference time. Additional values of masks size (16x16, 128x128 ?) and a discussion about the trade-off and adaptation to the dataset of use would be an additional valuable contribution.

The section 4.2.3 about generalization across modality shows promising results, but it misses the illustration of the latent space projections and some segmentation maps outputs as in Fig 2. They could be added in Appendix for instance.

The (potentially true) hypothesis on why the other methods yield lower results than the proposed ones are not supported by experiments, please adopt a less confident tone when taking the discussion on these hypothesis (can  => could, etc ...).

Description of Fig 4 and Fig 5 are not precise enough: please add letters on the image samples and attention maps and link their description in the legend accordingly. Guide the reader to better understand what the attention maps is supposed to look at in each case.
Fig 4 : why not displaying the results of the 64x64 masking as the results are better ?

The preliminary results obtained on cross-modality are promising but from the T2 (learn) to T1 (infer) the domain shift seems to be better managed by autoDDPM, can you comment on this ?

The post-processing does not seem to be applied to the outputs of autoDDPM (Fig.2 (c)), same for the Atropos segmentation mask integration (post processing described on page 5). Please also apply the post-processing steps to the autoDDPM (and other concurrent methods) outputs and recompute the metrics accordingly.

**Detailed Comments:**

page 2, last paragraph, "DDPM" : undefined acronym for Denoising Diffusion Probabilistic Models. Check that all acronyms are previously defined before use.

Double check the paper spelling, there are several typos or misleading formulations (for instance "Psedo" in legend of Figure 1, the title of section 4.2.2 can let the reader think you are masking the latent space, ...).

**Justification Of Final Rating:**

I appreciate the clarity improvements brought by the edits the authors did on the paper. The post-processing was an obvious topic of discussion for this new version of the paper. Despite a valid answer to my question about this post-processing "imbalance" to present a fair comparison of the compared methods, the authors do not demonstrate that it is indeed their mainly claimed contribution ("utilize a Swin Transformer-based Masked Auto Encoder (AnoswinMAE) for detecting and localizing pathologies in an unsupervised manner") and not the post-processing that brings the positive performance gap for the proposed method... comparing with/without the post-processing would have been a very convincing answer. Grading unchanged.

Figure 5 and Figure 6 : Row(ii) and (iii) a=> Row (ii) and (iv)

**Justification Of The Preliminary Rating:**

By addressing the difficult task of unsupervised anomaly detection in humain brain MRI, the authors contribution presents very promising results with an improvement of the performance compared to very recent methods such as autoDDPM (Bercea et al. 2023) with a +7% DICE increase. The preliminary results obtained on cross-modality are promising, in particular from the T1 (learn) to T2 (infer) domain shift  setup.

**Questions To Address In The Rebuttal:**

The question to address were grouped in the Weaknesses section alongside with the points to treat.

Can you comment about the inference time that is longer for Ano-swinMAE (32x32) compared to its (64x64) version (Table. 1, page 6) ?

---

> ### Author Response · Authors · 2024-03-17
> **Addressing  The Rebuttal Questions**
>
> We sincerely thank the reviewer for acknowledging the strengths of our manuscript and for highlighting points that can improve the understanding of our manuscript. We have incorporated suggested changes in the manuscript and highlighted the changes. We have addressed the weaknesses and questions as follows:
>
>
> [1] The meta-parameter defining the mask shape is critical …
> > We acknowledge the importance of tuning the hyperparameter and analyzing the results at more number of mask sizes. We have performed experiments on MSLUB and BraTS21 dataset with mask size 128X128. The Dice score (%) obtained for MSLUB is 17.42%, and for BraTS21 it is 40.25%. The AUPRC score (%) obtained for MSLUB is 12.32%, and for BraTS21 it is 27.91% Hence comparing with mask size 64X64 we have a performance drop in both cases. The increase in mask size from 64 to 128 covers most of the regions of the brain which hinders the reconstruction capability. Additionally, with mask size 16X16 the Dice and AUPRC for MSLUB is 18.95% and 12.44% respectively, when comparing it with the results of 32X32, there is a drop in performance.
>  Hence the optimal mask size for MSLUB is 32X32, and for BraTS21 is 64X64, which shows that this hyperparameter is dependent on the size of the pathology we are dealing with.  However, the computational complexity increases with the decrease in mask size.
>
>
> [2] The section 4.2.3 about generalization across modality shows promising results…
> >  It would definitely be an important addition. We have added a figure (Figure 7) in the Appendix which shows the results in the cross-modality setup.
>
> [3] The (potentially true) hypothesis on why the other methods yield..
> > We completely agree with the comment of the reviewer. We have changed ‘can’ with ‘could’ and ‘may with ‘might’ in the Quantitative And Qualitative Analysis section where we stated our hypothesis.
>
> [4] Description of Fig 4 and Fig 5 are not precise enough…
> > Figures 4 and 5 have had their captions revised to enhance clarity. Additionally, a new section "Visualization of Attention Maps" has been included in the Appendix, providing detailed descriptions of the attention maps.
>
> [5] Fig 4 : why not displaying the results of the 64x64 masking...
> > We have added a Figure (Figure 6) in the Appendix displaying the results of 64x64 masking.
>
> [6] The preliminary results obtained on cross-modality are promising...
> > It is a valid concern and needs to be highlighted. T2-weighted MRI highlights pathologies like tumors with high water content or those surrounding edema. Hence pathologies mostly become more visually discriminatable in T2 modality. When, we are assessing the metrics for T2 while being trained on T1, we observe that the performance of Ano-swinMAE is better since the model is able to easily differentiate between healthy and pathological content. When the modalities are reversed (inferring on T1 while trained on T2) due to the lesser discriminative appearance of the pathology, the performance of the model drops. There is an analogy between the appearance of the pathology and the performance of the model since it attempts to fill the masked regions with contextual neighborhood information. We have supported our claim by adding an image in the Appendix (Figure 7). In the case of autoDDPM the trend is reversed since having a more evident appearance of pathology (as in T2), the chances to pass the pathological information through the noising process is higher. This degrades the performance scores.
> We have added this reasoning in SubSection 4.2.3 as “ Ano-swinMAE exhibits better performance when trained on T1 and evaluated on T2 since T2 enhances the pathological appearances in the images and it is easily discriminative. Whereas in the case of autoDDPM if the pathological information is evident, then it is present after the noising process.”
>
> [7] The post-processing does not seem to be applied ...
> > We thank the reviewer for highlighting this. We have evaluated all the baseline methods by adopting the existing implementations available in the official repositories, except for MAE which was adopted from a paper demonstrating its applicability in the classification of natural images.
>
> [8] Can you comment about the inference time that is longer...
> > We acknowledge the significance of reasoning out the difference in the inference time. The increase in inference time for 32x32 mask size is due to the increase in the number of masked images that has to be processed.
>
> [9] page 2, last paragraph, "DDPM" : undefined acronym...
> > It is a valid concern. We have added the definition of the acronym in the Related Work section.
>
>
> [10] Double check the paper spelling...
> > We definitely need to address these typos. We have replaced “Psedo” with “Pseudo”. We changed the title of section 4.2.2 as Analysing The Effect Of Masking in The Latent Space. Also added a line “ In Figure 3, we compare the latent space of masked and unmasked images.” in the description.

---

### Official Review · Reviewer_4BUX · 2024-02-29

**Confidence:** 4
**Preliminary Rating:** 3
**Recommendation:** Poster
**Final Rating:** 3.5

**Summary:**

Authors propose an unsupervised anomaly detection approach to detect pathological regions in brain MRI data. The approach is based a Swin-tranformer-based Masked Autoencoder model trained to reconstruct healthy images. At inference, pathological regions are detected by subtracting the reconstructed pseudo-healthy images and the input pathological image using a sliding window algorithm. The method is evaluated on three publicly available datasets, namely BraTS, MSLUB, and WMH and compared to other generative and autoencoding approaches.

**Strengths:**

- Overall, the paper is well-written, the organization is clear;
- Use of four different publicly available datasets to evaluate the method;
- Valuable boost in performance while training on one dataset and testing on two others, as well as demonstrating consistent generalizability results when evaluating on different MRI sequences.

**Weaknesses:**

- Similar related work, including masking strategies for anomaly detection, with diffusion models for medical images [A], with transformer models in industrial images [B, C], and for medical images, especially brain MRI data [D], are missing in reference to prior work and should be discussed by authors, how their approach compares to these;
- Methodology (and pre- and post-processing) misses some precise implementation details, which hamper reproducibility, as well as code is not available ("will be available soon").

[A] Iqbal et al. Unsupervised Anomaly Detection in Medical Images Using Masked Diffusion Model. Workshop MLMI. 2023
[B] Hu et al. Features Masked Auto-Encoder-Based Anomaly Detection in Process Industry. DDCLS. 2023.
[C] Yu et al. Image Anomaly Detection and Localization Using Masked Autoencoder. ICONIP. 2023.
[D] Georgesc. Masked Autoencoders for Unsupervised Anomaly Detection in Medical Images. Procedia Computer Science. 2023

**Detailed Comments:**

Major comments:
- Not sure the name of the proposed method is a correct keyword;
- Section 3: The described methodology is for 2D images, while brain MRI data is 3D, please clarify;
- Section 4: "For post-processing, morphological filters are applied" this needs to be more precise. What operator? What structural element? Also, do the comparison methods also incorporate such heavy post-processing?
- Section 4: It's not clear how the Atropos segmentation algorithm and results were used;
- Appendix figures are not discussed; please refer to and analyze these figures.

Minor comments:
- Section 4: "normalization to the [0,1] intensity range." linearly?

**Justification Of Final Rating:**

The rebuttal provided clarifications on my concerns regarding the pre-/post-processing and implementation details of the methodology, and are quite heavy.
However, the strengths/weaknesses balance remains similar to me. Hence the choice of slightly increasing my rating to borderline accept.

**Justification Of The Preliminary Rating:**

The paper is well-written and the organization is clear. Some references to and discussion of prior work, as well as methodological description, are missing. But overall, the evaluation seems robust and the approach demonstrates consistent results.

**Questions To Address In The Rebuttal:**

- Authors should better incorporate and discuss references to prior work using masking strategies for anomaly detection. This includes specifying the contributions of the proposed approach;
- The methodology description should be more precise, especially regarding implementation details of the pre- and post-processing, the sliding window, and the use of the segmentation model; code availability would highly help;
- Answer major comments.

**Special Issue:**

No

---

> ### Author Response · Authors · 2024-03-17
> **Addressing The Rebuttal Questions**
>
> We sincerely thank the reviewer for their insightful comments and for acknowledging the strength of the manuscript. We have incorporated suggested changes in the manuscript and highlighted the changes. We addressed the questions,  Major and Minor comments as follows:
>
> [1] Authors should better incorporate and discuss references to prior work …
>
> > We thank the reviewer for pointing to such relevant references. We have added the details of the cited papers in the Related Work section. The primary difference between the cited references and our paper is described below:
> (i) Addressing paper [A], in this work a diffusion model is used, and during the training phase, a mechanism to simulate pathology is introduced. The simulation is done by masking the original image and adding a frequency-altered version of the original image in the masked images. Their notion of introducing masks is to simulate pathologies unlike ours where we intend to mask the pathological image during inference.
> (ii) Addressing paper [B][C][D], these works employ transformer-based Masked Autoencoders (MAE) to generate a pseudo-normal counterpart of the normal image. Additionally, [D] integrates a pseudo-abnormal module to simulate pathology to train a classifier to discriminate between healthy and unhealthy. Their dependency on the secondary classifier clearly highlights that MAE independently is not suitable for pathology detection.
>
> [2] The methodology description should be more precise…
> > We acknowledge the necessity of adding these precise details. For pre-processing, we have used (i) skull-stripping (ii) Registration and (iii) Linear Normalisation (0-1). We have added the detailed post-processing steps and the sliding window method in the Appendix section (Appendix A.1) of the paper. We intend to make the code available upon acceptance. We have provided an empty repo link for traceability  (https://github.com/rashmi05pathak/Ano-swinMAE).
>
> [3] Not sure the name of the proposed method is a correct keyword;
> > It is a valid concern. We  Ano-swinMAE refers to the task of dealing with Anomaly detection using a swin transformer-based Masked Autoencoder.
>
>
> [4]Section 3: The described methodology is for 2D images, while brain MRI…
> > It is a valid concern. We are working with 2d slices of the 3d brain mri.
>
> [5] Section 4: "For post-processing, morphological filters are applied" ..
> > We thank the reviewer for highlighting this. We have added the detailed post-processing steps and the sliding window method in the Appendix section (Appendix A.1) of the paper. We have done post-processing based on morphological operations and connected component analysis which is generally adopted for Unsupervised brain Anomaly Detection. The additional component in our post-processing step is adding information from Atropos segmentation which only contributes to refining the boundary information. We have included a Figure in the Appendix that shows the boundary refinement due to the addition of Atropos segmentation. We have evaluated all the baseline methods by adopting the existing implementations available.
>
> [6] Section 4: It's not clear how the Atropos segmentation …
> > It is a valid concern. We have added the detailed post-processing steps and the sliding window method in the Appendix section (Appendix A.1) of the paper.
>
> [7] Appendix figures are not discussed…
> > We thank the reviewer for going through the appendix. We have added a section in the Appendix (A.2.) which describes the information presented in the Figures.
>
> [8] Section 4: "normalization to the [0,1] intensity range." linearly?
> > Yes we have performed a linear intensity normalization.

---

> > ### Comment · Reviewer_4BUX · 2024-03-22
> > **Response to rebuttal**
> >
> > Thank you for your detailed answer and the modifications made to the manuscript.
> > Clarifications were made, especially on the pre-/post-processing parts.

---

### Official Review · Reviewer_QR1v · 2024-02-29

**Confidence:** 4
**Preliminary Rating:** 3
**Recommendation:** Poster
**Final Rating:** 4

**Summary:**

In this paper, the authors developp a method based on masked auto-encoders with swin transformers (Dai et al 2023), for anomaly detection in brain MRI. The method is tested on public datasets Brats, MSLUB and WMH, and also in cross-modality (e.g. train on T1 test on T2). The authors provide additional experiments analysing the latent space with/without masking that strengthen their about the usefulness of masking for learning the healthy data manifold.

**Strengths:**

1) The method seems novel
2) The experiments are well conducted and clear
3) Datasets used are public
4) Method is tested on various datasets and in cross-modality
5) Additional latent space analysis is carried to strengthen the authors claims

**Weaknesses:**

1) The reviewer felt like a lot of claims (pinned bellow) made in the discussion were vague/unprecise. Additional efforts have to be made for clarity of the discussion and to guide the reader towards the main claims of this paper.

**Detailed Comments:**

- "have proven to be effective during both training and inference." : can the authors add some references (used elsewhere in the text)
- Figure 1 : "psedo" -> "pseudo"
- "x to x̂, both of which belong to a healthy cohort." : I would not say x_hat belongs to a "healthy cohort", as it is a reconstruction, a synthetic reconstruction of a healthy patient.
- "The token of a masked patch is allowed to pass through the model such the model" --> such as ?
- "Area Under the Precision Accuracy curve (AUPRC)"  -> Recall instead of Precision
- "The code for our proposed method will be available soon." : please provide the link in the rebuttal, even if the repo is empty, for tracability.
- "The pathology segmentation masks produced by autoDDPM, MAE, and our method for BraTS21 T2 data are illustrated in Figure 5." I think the authors meant Figure 2.
- "The autoDDPM model tends to generate pathologies" --> detection, or other wording
- Figure 2 : The reviewer is unsure but thinks it would be more intuitive to have the TP in green.
- "In the case of WMH Flair data, our model has comparative results to the baseline." : I would add "and slightly worse AUPRC"
- "tending to form differentiable clusters" : I would not use the term "differentiable" which has a precise mathematical meaning
- "which further extends the scope of transformers in the field of medical imaging" : I would add "transformer usage"

**Justification Of Final Rating:**

The comments were all addressed and improved the manuscript overall clarity, which was lacking in the first version. I thus change my grading from borderline to weak accept, as I think this is of interest to present to the conference.

**Justification Of The Preliminary Rating:**

The reviewer thinks that the paper method's is novel enough (but is curious to see other reviewers comments on that matter), the experiments are well conducted, on public databases. However the reviewer felt like a lot of claims (pinned above) were not grounded or unprecise and would like the authors to clarify and precise them. If done the reviewer would happily switch to weak accept or more.

**Questions To Address In The Rebuttal:**

- "However, ViT-based models lack (Bietti and Mairal,2019) inductive biases which cannot trade with global context since inductive biases are crucial for locality information." : can the authors precise what inductive bias they are thinking about ?
- "and image reconstruction tradeoffs used for UAD tasks are (Behrendt et al., 2023) and (Bercea et al., 2023). This tradeoff often limits the applicability of the models among diverse pathological data" : can the authors precise what tradeoff they are thinking about ?
- The method seem to be very similar (SwinMAE) to the method of Dai et al 2023, can the authors list explicitely the difference with this method ?
- For the post-processing (morpho maths, connected components, etc.) could the authors provide the exact operations in appendix for reproducibility ?
- The atropos framework is used for coarse segmentation, which is then said to "refine the anomaly map", can the authors give visual exemples or precise what type of refining is done with this segmentation ? Moreover, on Figure 1 the structures (mixture components) are ranked in such a way that 1 component has higher intensity than another, etc. How is this ordering done ?
- Can the authors precise if the performances for the other competiting methods are taken from the literature or are the methods reimplemented ?
- The AU PR is used as a metric, the baseline for this metric is not 0.5 as for the AU ROC, can the authors provide the baseline (obtained with a random classifier) for the datasets used ?
- Is the inference time per volume or for the whole dataset ? Or else ?
- The reviewer found that the whole block beginning with "MAE could not achieve performance" and ending with "resulting in low-fidelity images and an increased occurrence of false positives." was very unclear. Can the authors detail each sentence, precise what they mean exactly ? The interpretation seems interesting but the reviewer feels like as is it is not understandable.
- "Notably, the integration of superpixel information enhances the precision of anomaly boundaries." : can the authors precise what they meant ? The term "superpixel" first appears in this sentence.
- "Each of the ellipses formed from the covariance matrix of each cluster mostly contains projected points from one of the datasets (IXI or BraTS21)" : I would change "mostly" to "rhoughly", as one of the ellipse containt both purple and yellow dots.
- Table 2 : with which patch size are the results obtained ?
- Can the authors comment on the fact when training on IXI and testing on brats for cross-modality (table 2), the results are very different if T1 -> T2 or T2 -> T1 ?
- "Consequently, this allows the latent representational space to encode diverse semantics, unlike modelling capabilities like GAN and VAE, which restrict the latent space." : again, can the authors precise what they meant ? What is restricted in GAN and VAE that is not in Transformers ?
- The main text, and the legend of the Figure 4 and 5 have to explain what is done in these figures. As is it is not understable, although it looks interesting.

**Special Issue:**

No

---

> ### Author Response · Authors · 2024-03-17
> **Addressing the Rebuttal Questions Set1**
>
> First of all, we thank the reviewer for their insightful and precise comments which will definitely elevate the quality of the paper. We have addressed the Detailed Comments and highlighted them in the manuscript. The details of each of the comments are discussed below:
>
>
> [1] "However, ViT-based models lack (Bietti and Mairal,2019) inductive biases…
> > We acknowledge that precisely mentioning “inductive bias” is necessary. Here inductive bias refers to the modeling of neighborhood/ locality structures which is an implicit property of a Convolutional Neural Network (CNN). The convolution operations in CNNs extract local features from the neighbor pixels within the receptive field determined by the kernel size. Inductive Bias follows the intuition that local pixels are more likely to be correlated in images. We have cited an additional reference in the paper and rephrased  “However, ViT-based models lack (Bietti and Mairal, 2019) inductive biases which cannot trade with global context since inductive biases are crucial for locality information.”  in the following manner -- “However, ViT-based models lack (Xu et al., 2021)(Bietti and Mairal, 2019)inductive biases in modeling local visual structure which cannot be traded with global context since inductive biases for locality information is crucial as we can see in the case of CNN”.
>
>
> [2] "and image reconstruction tradeoffs used for UAD tasks …
> > We understand that clarity is required in this phrase. To address this, we have added this line in the paper “In the DDPM-based approach, there is a tradeoff between preserving crucial information about healthy tissues and efficiently eliminating anomalies due to the inherent noise addition strategy”. With the increase in noising time steps, the chances of removal of pathological information increases while there is a compensation of the amount of healthy region that will be reconstructed back.
>
> [3] The method seem to be very similar (SwinMAE)...
> > It is a valid concern. Our architectural proposition has similarities to Dai et al 2023, and their proposition is to derive a downstream task out of the pre-trained swinMAE network. whereas we propose to use the model's implicit ability to fill masked patches with information from the learnt distribution. Consequently, we propose tackling the anomaly-to-pseudo-healthy generation task by incorporating this inherent capability of the model.
>
>
> [4] For the post-processing ..
> > We completely understand the necessity of adding these details. We have added the detailed post-processing steps and the sliding window method in the Appendix section (Appendix A.1) of the paper.
>
> [5] The atropos framework is used for coarse segmentation ...
> > We acknowledge the importance of highlighting the significance of adding Atropos framework. We have included the images in the Appendix section. The Atropos framework is used to refine the boundaries of pathologies by incorporating the segmented information from the original MRI slice. In Figure 1, we have not adopted any ranking based on intensities, whereas in Figure 2, the ranking is done based on the visual intensity appearance of the pathological regions in T2 imaging modality compared to other healthy bright regions (CSF, sulci).
>
> [6] Can the authors precise if the performances for the other competiting methods …
> > We understand the significance of highlighting this. We have reimplemented all other baseline methods.
>
> [7] The AU PR is used as a metric, the baseline for this metric is not 0.5…
> > Yes the AUPRC metric does not have a fixed baseline but rather depends on the fraction of positives present within a dataset. We have used a random classifier for BraTS21 dataset and inferred the following: We have 3-4% of the pixels/slice (average) with pathologies in the entire dataset, evaluating with a random classifier the AUPRC scores range from 0.02 to 0.2. It is to be noted that the randomness of the classifier introduces a range of scores.
>
> [8] Is the inference time per volume or for the whole dataset ? Or else ?
> > It is a valid concern, we have added (per MRI slice) in Table 1.
>
> [9] The reviewer found that the whole block beginning with "MAE could not achieve performance" ..
> > We acknowledge that the explanation should be more clear. We have replaced “MAE could not achieve performance increments due to its dependence on a large volume of data for convergence, generating images with relatively lower fidelity.” with “ MAE could not achieve performance increments because of its reliance on a large amount of data, leading to lesser image fidelity compared to Ano-swinMAE”. The MAE model is dependent on a large amount of data, when we train it with lesser data (as we are using medical images) the model is not able to reconstruct good-quality images.

---

> ### Author Response · Authors · 2024-03-17
> **Addressing The Rebuttal Questions Set2**
>
> [10] The term "superpixel" first appears in this sentence.
> > Yes, indeed the term appears for the first time. We meant to denote that the accumulation of similar intensity with Atropos segmentation enhances the precision of anomaly boundaries. We have rephrased “Notably, the integration of superpixel information enhances the precision of anomaly boundaries.” with “Additionally, the integration of Atropos-based segmentation information enhances the precision of anomaly boundaries” in the paper.
>
> [11] I would change "mostly" to "rhoughly", as one of the ellipse contain both purple and yellow dots.
> > Yes we understand the concern, We have addressed this by changing “mostly” to “approximately” in the paper.
>
> [12] Table 2 : with which patch size are the results obtained ?
> > It is a valid point and needs to be mentioned in the description. We have added it in the description of Table 2. The patch size we have performed generalization with is 64 X 64.
>
> [13] Can the authors comment on the fact when training on IXI and testing on brats for cross-modality …
> > It is a valid concern and needs to be highlighted. T2-weighted MRI highlights pathologies like tumors with high water content or those surrounding edema. Hence pathologies mostly become more visually discriminatable in T2 modality. When, we are assessing the metrics for T2 while being trained on T1, we observe that the performance of Ano-swinMAE is better since the model is able to easily differentiate between healthy and pathological content. When the modalities are reversed (inferring on T1 while trained on T2) due to the lesser discriminative appearance of the pathology, the performance of the model drops. There is an analogy between the appearance of the pathology and the performance of the model since it attempts to fill the masked regions with contextual neighborhood information. We have supported our claim by adding an image in the Appendix (Figure 7). In the case of autoDDPM the trend is reversed since having a more evident appearance of pathology (as in T2), the chances to pass the pathological information through the noising process is higher. This degrades the performance scores.
> We have added this reasoning in SubSection 4.2.3 as “ Ano-swinMAE exhibits better performance when trained on T1 and evaluated on T2 since T2 enhances the pathological appearances in the images and it is easily discriminatable. Whereas in the case of autoDDPM if the pathological information is evident then is present after the noising process.”
>
> [14]  What is restricted in GAN and VAE that is not in Transformers ?
> > We acknowledge the necessity of mentioning the details about restriction in latent space. We have rephrased the previous content with the following and added it to the manuscript --- Consequently, this allows the latent representational space to encode diverse semantics, unlike modeling capabilities like GAN and VAE, which constrain the latent space to follow a prior standard Gaussian distribution.
>
> [15] The main text, and the legend of the Figure 4 and 5…
> > We thank the reviewer for going through the appendix. We have added a section in the Appendix (A.2.) which describes the information presented in the Figures.

---

> ### Author Response · Authors · 2024-03-17
> **Addressing The Detailed Comments**
>
> We have addressed the detailed comments of the reviewer as follows:
>
> [1] "have proven to be effective during both training and inference."..
> > We have added two references (1) “Autoencoders for unsupervised anomaly segmentation in brain MR images: A comparative study” and (2) “VAE-based Deep SVDD for anomaly detection” in the manuscript, which are consolidated review articles on Autoencoder-based unsupervised anomaly detection.
>
> [2] Figure 1: "psedo" -> "pseudo"
> > We have corrected the typo in Figure 1.
>
> [3] "x to x̂, both of which belong to a healthy cohort."..
> > We have addressed this by referring to  x̂  as the reconstructed image instead of claiming that it belongs to the healthy cohort.
>
> [4] "The token of a masked patch is allowed to pass through the model such the model" --> such as ?
> > Here we have added a connective word “such that” instead of “such”.
>
> [5] "Area Under the Precision Accuracy curve (AUPRC)" -> Recall instead of Precision
> > Thanks for highlighting this typo, we have changed Accuracy to Recall.
>
> [6] "The code for our proposed method will be available soon." …
> > We have provided the link for traceability https://github.com/rashmi05pathak/Ano-swinMAE
>
> [7] "The pathology segmentation masks produced by autoDDPM, MAE, and our method for BraTS21 T2 data are illustrated in Figure 5." I think the authors meant Figure 2.
>  > The overlapping of Figure numbers(2 & 5) is due to the same label assignment (\label{} in latex) to the figures. We have changed the label assignment of the figures.
>
> [8] "The autoDDPM model tends to generate pathologies" --> detection, or other wording
> > It is a valid concern. We have changed the word “generate pathologies” to “reconstruct traces of pathologies”.
>
> [9] Figure 2: The reviewer is unsure but thinks it would be more intuitive to have the TP in green.
> > We understand the concern and have changed TP to green, FP to blue and FN to red.
>
> [10] "In the case of WMH Flair data, our model has comparative results to the baseline." : I would add "and slightly worse AUPRC"
> > We understand the concern that comparative doesn't exactly quantify the performance. We have added “slightly inferior” instead of comparative.
>
> [11] "tending to form differentiable clusters": I would not use the term "differentiable"..
> > We have replaced the term “differentiable” with “separable”.
>
> [12] "which further extends the scope of transformers in the field of medical imaging": I would add "transformer usage"
> > We have replaced “the scope of transformers” with "transformer usage".

---

> > ### Comment · Reviewer_QR1v · 2024-03-26
> >
> > The reviewer thanks the authors for the clarifications and responses made to the concerns.

---

### Meta-Review · Area_Chair_GwPY · 2024-04-05

**Recommendation:** Accept (Poster)
**Confidence:** 3

**Metareview:**

All reviewers found the proposed method to be novel, positively evaluated the fact that is based on public data, and the results promising.

---

### Decision · Program_Chairs · 2024-04-05

Accept (Poster)